# Transcriptomic Responses to Thermal Stress and Varied Phosphorus Conditions in *Fugacium kawagutii*

**DOI:** 10.3390/microorganisms7040096

**Published:** 2019-04-02

**Authors:** Senjie Lin, Liying Yu, Huan Zhang

**Affiliations:** 1Department of Marine Sciences, University of Connecticut, Groton, CT 06340, USA; 2State Key Laboratory of Marine Environmental Science, Xiamen University, Xiamen 361102, Fujian, China; yly20070567@126.com; 3Department of Marine Sciences, University of Connecticut, Groton, CT 06340, USA; huan.zhang@uconn.edu

**Keywords:** *Fugacium kawagutii*, corals, thermal stress, nutrient limitation, phosphate, dissolved organic phosphorus (DOP), transcriptomics

## Abstract

Coral reef-associated Symbiodiniaceae live in tropical and oligotrophic environments and are prone to heat and nutrient stress. How their metabolic pathways respond to pulses of warming and phosphorus (P) depletion is underexplored. Here, we conducted RNA-seq analysis to investigate transcriptomic responses to thermal stress, phosphate deprivation, and organic phosphorus (OP) replacement in *Fugacium kawagutii*. Using dual-algorithm (edgeR and NOIseq) to remedy the problem of no replicates, we conservatively found 357 differentially expressed genes (DEGs) under heat stress, potentially regulating cell wall modulation and the transport of iron, oxygen, and major nutrients. About 396 DEGs were detected under P deprivation and 671 under OP utilization, both mostly up-regulated and potentially involved in photosystem and defensome, despite different KEGG pathway enrichments. Additionally, we identified 221 genes that showed relatively stable expression levels across all conditions (likely core genes), mostly catalytic and binding proteins. This study reveals a wide range of, and in many cases previously unrecognized, molecular mechanisms in *F. kawagutii* to cope with heat stress and phosphorus-deficiency stress. Their quantitative expression dynamics, however, requires further verification with triplicated experiments, and the data reported here only provide clues for generating testable hypotheses about molecular mechanisms underpinning responses and adaptation in *F. kawagutii* to temperature and nutrient stresses.

## 1. Introduction

Dinoflagellates are known to have several major roles in the ocean: important primary producers, the greatest contributors of harmful algal blooms and marine biotoxins, and essential endosymbionts of reef building corals and some other invertebrates. The coral endosymbionts come from the family of Symbiodiniaceae [1], which are classified into nine clades (A–I) based on their genetic differences despite little morphological variance [2], some of which have just begun to be recognized as different genera [3]. 

Among the nine clades of Symbiodiniaceae, clade E is exclusively free-living, while the others contain endosymbionts. Coral symbionts mostly fall in clades A, B, C, D, and F [4]. Our understanding on the coral–Symbiodiniaceae relationship has evolved from the earlier “one coral–one genotype of Symbiodiniaceae” to “one coral–one dominant genotype of Symbiodiniaceae” with the discoveries that the endosymbiont assemblage contains multiple genotypes [1,5,6]. The dominant and minor genotypes can shuffle in the process of coral bleaching [7,8]. Coral bleaching, the increasingly widespread and severe coral-degrading phenomenon, is due to the expulsion of Symbiodiniaceae as a consequence of environmental stress [9]. Living in the tropical oligotrophic oceanic environment, the limitation of nutrients, such as phosphorus (P) [10], in conjunction with episodes of El Nino and undergoing global warming, can exacerbate coral bleaching. 

It has been reported that thermal stress induces inactivation of photosystem II (PSII) in Symbiodiniaceae species by damaging light-harvesting proteins [11]. Studies have also shown that symbiosis with different genotypes of Symbiodiniaceae can lead to differential susceptibility of the coral to thermal stress. For instance, Symbiodiniaceae of putatively thermotolerant type D2 and the more susceptible type C3K showed markedly different gene expression profiles, especially for heat shock proteins and chloroplast membrane components within 3 days of exposure to elevated thermal exposure [12]. Some studies reported a depression of growth rates and photosynthesis at high temperatures in clades A and B, but not in clades D and F [13]. At an elevated temperature of 31 °C, clade F showed a transcriptional response in 37.01% of its 23,654 total detected unigenes, among which 2.78% exhibited ≥2-fold changes in expression, and these responsive genes encoded antioxidant and molecular chaperones, cellular components, and other functions [14]. More studies like this on different types of Symbiodiniaceae species can help us better understand how the symbiont community responds to environmental stress. Besides, an investigation of the response to nutrient stress is also important for corals and Symbiodiniaceae species as they typically live in oligotrophic environments. Phosphorus is an essential nutrient for algae, but its directly bioavailable form (primarily dissolved inorganic phosphate or DIP) is often limited in various parts of the global ocean [15,16]. Natural and human activities introduce dissolved organic P (DOP) into coastal waters, providing an alternative P source [17]. No study has specifically addressed the molecular response to P stress and the replacement of DOP for phosphate in Symbiodiniaceae species.

We conducted a transcriptomic study on *Fugacium kawagutii* treated with heat stress (30 °C), P deficiency, and DOP replacement. *F. kawagutii* belongs to clade F and was originally isolated from the scleractinian coral, *Montipora verrucosa*, in Hawaii, where the ambient temperature is about 25 °C [18], but this specific genotype has only been occasionally found in subsequent studies (in *Pocillopora damicornis* in Heron Island [19] and in Hong Kong [Lin unpublished data], both in the Pacific). This raises a question of whether this is like the thermal resistant clade D genotypes that are rare under normal conditions, but resistant to stress due to a physiological tradeoff [20,21,22]. A recent genome study indicated that *F. kawagutii* possesses microRNA gene regulatory machinery that potentially targets heat shock proteins, and that the species has an expanded gene repertoire of stress responses [23]. Besides, this species shows highly duplicated nutrient transporters, including phosphate transporters and alkaline phosphatases as well as acid phosphatases, that are potentially helpful for utilizing DOP (Table S21 in Lin et al. [23]). Our transcriptomic analyses to identify differentially expressed genes reported here provide insights into responses in *F. kawagutii* to both temperature and P nutrient variations.

## 2. Materials and Methods 

### 2.1. F. kawagutii Cultures, Sampling, and RNA Sequencing

*F. kawagutii* strain CCMP2468 was obtained from the National Center for Marine Algae and Microbiota (NCMA) in the Bigelow Laboratory of Ocean Science (Boothbay Harbor, ME, USA). It was maintained in L1 medium prepared from natural seawater under an illumination of ~200 μE m^−2^ s^−1^ with a 14:10 light dark cycle and at a temperature of 25 °C. For the experiment, 1 L of the cells in the exponential growth stage were divided into 4 bottles and collected by centrifugation at 3000× *g*, 25 °C for 15 min to remove the old cultural medium, and re-suspended with different growth media and divided into 4 groups (SymkaSL1–4), each in triplicate. SymkaSL1 and SymkaSL2 were grown in artificial seawater enriched with normal standard L1 medium (nutrient complete), SymkaSL3 in artificial seawater-based L1 medium with deprived phosphate (P-), and SymkaSL4 in artificial seawater-based L1 medium with DIP replaced by the DOP glycerol-3-phosphate (Gro3P) at the equivalent concentration (36.2 µM). Each subsample was washed with the corresponding cultural medium 3 times to remove trace amounts of old culture medium and transferred to triplicated 150-mL bottles. SymkaSL1 triplicate bottles were transferred to 30 °C for thermal stress treatment, while SymkaSL2–4 triplicate groups were kept at 25 °C. After one-week incubation under those conditions, cells were harvested for RNA extraction. 

RNA was isolated using Trizol reagent (Life Technologies, Grand Island, NY, USA) coupled with the QIAGEN RNeasy kit (QIAGEN Inc., Germantown, MD, USA) according to Zhang et al. [24]. The RNA samples were subjected to RNA-seq in 2 × 50 bp paired-end format at the National Center for Genome Resources under Marine Microbial Eukaryote Transcriptome Sequencing Project (MMETSP) [25]. Before sequencing, RNA extracts from the triplicate cultures were pooled at equal RNA quantity (due to high cost of sequencing in 2011). Raw data was uploaded to NCBI under the accession numbers, SRR1300302, SRR1300303, SRR1300304, and SRR1300305.

### 2.2. Data Preprocessing

The genome reference datasets of *F. kawagutii* were downloaded from http://web.malab.cn/symka_new/index.jsp for transcriptomic reads mapping. Cutadapt [26] was used to remove adaptor sequences with parameters of “-e 0.05 --overlap 25 --discard-trimmed -m 20” for each end of fastq data separately. The trimmed clean reads were evaluated with FastQC to check quality. Multiqc v1.3.dev0 (https://multiqc.info/) software was used to integrate FastQC reports.

### 2.3. Reads Counting

Clean reads were aligned to the genome sequences by HISAT2 [27] with default parameters. The generated sequence alignment map (SAM) was then converted into its binary format, BAM, and sorted using SAMtools [28]. Mapping quality was evaluated using Qualimap (bamqc) [29] based on BAM file. Read summarization was analyzed with featureCounts, and multi-mapping reads were not counted (parameters of -p -C). Read counts were normalized as fragments per kilo-base of exon model per million mapped fragments (FPKM). For each pair of comparison, to avoid inflation of differential expression by low-expression genes, genes with read counts per million (CPM) < 10 in both samples, or not detected in one sample while CPM < 20 in the other sample, were excluded. The remaining genes (herein named actively expressed genes or AEG) were subjected to further statistics analyses.

### 2.4. Identification of Expressed Core Genes

Commonly as well as highly expressed genes in *F. kawagutii* grown in all four conditions were identified as core genes. Several steps were used to accomplish this. Firstly, from the AEG, genes commonly detected in four samples were identified (AEG-C). Secondly, for each gene in SGS-C, the average FPKM and coefficient of variance (CV) in the four samples were calculated. Finally, genes showing expression of ≥50th percentile average FPKM and low CV of ≤0.1 were collected and named cores genes.

### 2.5. Differential Gene Expression Analysis

To generate differential gene expression profiles in the absence of biological triplicates (due to the pooling of the triplicate culture samples), two methods, edgeR v3.24.3 (http://bioconductor.org/packages/edgeR/) and NOIseq v2.26.1 (http://bioconductor.org/packages/NOISeq/) in R, were used in parallel and only the consistent results from both methods were used for further analysis. Both methods account for biological variability when samples have no replicates. edgeR determines DEGs using empirical Bayes estimation as well as exact tests based on negative binomial models [30], and is widely used for analyzing DEGs for non-replicated samples. The moderate biological coefficient of variation (BCV) of 0.2 was used to estimate dispersion. DEGs of edgeR were screened with threshold of FDR ≤ 0.05 and the absolute value of log2Ratio ≥ 1. NOIseq is a non-parametric approach consisting of NOISeq-real and NOISeq-sim. NOISeq-sim simulation of noise distribution in the absence of replication was optimized. The parameters of *q* = 0.9, pnr = 0.2, nss = 5, and *v* = 0.02 were set as previously suggested [31]. Only genes identified as DEGs by both methods were classified as true DEGs. Heatmap was used to display DEGs with value of log2(RPKM + 1) transformation by pheatmap v1.6.0 in R.

### 2.6. Gene Ontology and KEGG Functional Enrichment 

To optimize the gene ontology (GO) annotation rate, the InterProscan tool was used to scan protein signatures of the newest updated databases (version 5.27-66.0, lookup_service_5.27-66.0, http://www.ebi.ac.uk/interpro/interproscan.html) against *F. kawagutii* genome protein sequences. GO annotation results were visualized by WEGO 2.0 [32]. Kyoto Encyclopedia of Genes and Genomes orthology (KO) information was grabbed from *F. kawagutii* genome KEGG annotation information. We used significant DEGs as a foreground to perform GO or KEGG functional enrichment analyses with the ClusterProfiler package [33]. And enrichment visualization was displayed with the enrichplot package [34]. Because of the largely unexplored genome of Symbiodiniaceae species, the annotation rate of GO and KEGG were generally low.

## 3. Results

### 3.1. Overall Differential Gene Expression Profile

The RNA-seq yielded 550 to 1150 Mbp for the four culture conditions: SymkaSL1 (thermal stress at 30 °C), SymkaSL2 (control), SymkaSL3 (phosphate deprivation), and SymkaSL4 (Gro3P replacement). Approximately 60% of each dataset was successfully mapped to the genome of this species (Table 1). In total, 44.72% of the genome-predicted genes (36,850) were covered by all the transcriptomes combined. Each of the three treatments was compared with the control. To avoid inflation of DEG numbers by genes with low-expression genes, sequences that were expressed at a low level (<10 CPM) in both two samples, or not expressed in one sample and expressed <20 CPM in the other, were excluded from the DEG analysis. This filtering resulted in a similar set of genes (~8000, actively expressed genes or AEG) for DEG analysis for each treatment (Table 2). The edgeR and NOIseq methods accounting for biological variability in our samples without replication were simultaneously used. Of this AEG dataset, while edgeR identified 5.57%–13.33% as DEGs with statistical significance (FDR ≤ 0.05 and fold change ≥ 2), NOIseq identified 13.71%–56.47% as DEGs with statistical significance (*q* = 0.9). The number of genes identified as DEG by both NOIseq and edgeR was 4.42%–8.05% of the ~8000 genes. Further functional analysis was based on this smaller set of statistically significant DEGs.

From the AEG set, 10,857 genes were found to be commonly expressed genes in all the four culture conditions (named AEG-C gene set) (Table 2). Of the AEG-C set, 221 (2.04%) genes showed no significant differential expression based on our criteria. Additionally, these are considered “core” genes (CORE) in this species. The average expression of the core genes was 584 FPKM, the lowest expression was 26 FPKM while the highest expression was up to 13,450 FPKM. Of these 221 CORE genes, 108 (48.87%) were functionally annotatable (Appendix A). Among these annotatable CORE genes, the most highly expressed gene encodes 14-3-3 protein, and the second encodes adenosine diphosphate (ADP) ribosylation factor. Functions of CORE genes were confirmed by GO annotation. The 84 GO annotatable genes were distributed in two subcategories of cellular component, six subcategories in molecular functions, and three subcategories in biological processes (Figure 1). Two of the sub-categories of molecular functions were highly enriched: Catalytic activity and binding. Included in the catalytic activity subcategory were oxidoreductase, hydrolase, and transferase activities. There were eight types of binding, including organic cyclic and heterocyclic compound binding, protein binding, and ion binding (Figure 1).

### 3.2. Functional Distribution of DEGs Responding to Heat Stress

Under thermal stress, *F. kawagutii* showed 357 (4.42%) DEGs, with over two-thirds being up-regulated and nearly one-third being down-regulated (Table 2). Of these DEGs, 171 genes (47.90%) were functionally annotatable (Appendix A). As expected, expression of heat shock proteins (HSPs), including HSP40, HSP70, and HSP90, and chaperonin Cpn60 was strongly elevated during heat stress. Furthermore, the abiotic stress-induced glutathione s-transferase was transcriptionally promoted by thermal stress. A phytoglobin (Skav228962), a plant counterpart of animal hemoglobin involved in binding and/or transporting oxygen [35], previously suggested as a stress biomarker [36], was also up-regulated. Reversely, the conjugative protein genes, including the ATP aldo/keto reductase family and sulfotransferase, were depressed. Photosystem I reaction center subunit IV and two iron permease (FTR1) genes involved in iron uptake were also markedly suppressed (by 5–10 fold). In transport activities, amino acid transporter, formate/nitrite transporter, ion transport protein, and p-type ATPase transporter were highly up-regulated. In contrast, choline transporter-like protein, nucleotide-sugar transporter, and ATP binding cassette (ABC) transporters were down-regulated. In the signal transduction process, most genes encoding protein kinases, ATPase, and polycystin 2 showed elevated expression under heat stress. Meanwhile, GO enrichment analysis indicated that three GO terms were significantly enriched (Figure 2). The carbohydrate metabolic process was one of them, which consisted of a down-regulated alpha-D-phosphohexomutase, three up-regulated glycoside hydrolases, and an up-regulated fructose-bisphosphate aldolase. Iron binding activity was potentially promoted as inositol oxygenases and cytochrome c/P450 genes were significantly up-regulated. In addition, cytoplasm was enriched with markedly up-regulated inositol oxygenases, inorganic pyrophosphatase, and protein CfxQ. Interestingly, a protocaderin fat gene was highly expressed and induced by heat stress, so was cytokinin riboside 5’-monophosphate phosphoribohydrolase (Figure 2). Additionally, as duplicated gene families reported in the *F. kawagutii* genome [23] were up-regulated, including most of the ankyrin repeat (AR)-, F-box, and FNIP repeat- and pentatricopeptide repeat (PPR)-containing proteins, and EF-hand domain-containing calcium-binding proteins, Zinc finger (ZnF) proteins, glycoside hydrolases, the regulator of chromosome condensation (RCC1), and glycosyl transferases (Appendix A).

### 3.3. Functional Distribution of DEGs Responding to P Stress

P stress affected 396 genes (4.73%) based on DEG analysis (Table 2). More than 80% of these DEGs had significantly higher expression levels than the control whereas only 64 showed down-regulation based on our analysis method. Of the 396 DEGs, 231 (58.3%) were functionally annotatable (Appendix A). Among the up-regulated were genes coding for proteins involved in phosphate exchange between chloroplast and cytoplasm, triose phosphate/phosphate translocator (TPT) [37], and five phosphatases potentially involved in DOP metabolism and utilization, including alkaline phosphatase (AP), phosphoserine phosphatase, phosphoglycolate phosphatase, protein phosphatase 2C, and metal-dependent phosphohydrolase. 

Differential gene expression was richly represented in photosynthetic apparatus under the influence of P stress: Photosystem II (PSII) light harvesting complex proteins and chlorophyll a-c binding protein, photosystem I (PSI) reaction center subunit IV, and PsaD as well as electron-transfer proteins in the photosynthetic electron transfer chain of flavodoxin and ferredoxin. Under P stress, a total of 11 differentially expressed PSII chlorophyll a-c binding protein genes were up-regulated, and two light harvesting complex protein genes were down-regulated. Eight of the 11 PSII chlorophyll a-c binding protein genes were enriched in the GO term of photosynthesis light harvesting (Figure 3). The results were consistent with changes in gene expression of *Prymnesium parvum* induced by nitrogen and phosphorus limitation [38]. Two PSI PsaD genes were up-regulated, but the PSI reaction center subunit IV decreased to an undetectable level in the P-deprived cultures. GO enrichment showed that these three genes were concentrated in the photosystem I reaction center (Figure 3). Interestingly, the highly expressed flavodoxin was decreased while ferredoxin was enhanced under P stress; flavodoxin is known to be induced under iron limitation to replace its iron-containing functional equivalent, ferredoxin, in diatoms and some other algae [39,40,41,42]. 

P plays an important role in the production of ATP, NADH, and NADPH; thus, energy related genes of six DEGs were explored. They included three ATPase genes, a p-type ATPase transporter, glyceraldehyde 3-phosphate dehydrogenase, and geranylgeranyl diphosphate reductase. Among them, five were up-regulated while ATPase subunit C was down-regulated under P stress. Besides, oxidoreductase activity and heme binding terms were enriched, mostly comprised of up-regulated haem peroxidase, globin, cytochrome b5, and Please define this term if appropriate. (Figure 3). 

Under P deprived conditions, several stress responsive genes were observed (Figure 3). The antioxidant protein genes encoding superoxide dismutase were highly expressed and induced. Two oxidative genes, flavin-containing monooxygenases and the major facilitator superfamily MFS-1, were induced under P-stress condition. Also, heat shock proteins, DnaJ and Hsp90, were differentially expressed. Several genes of protein kinases were up-regulated. Among these, a serine/threonine-protein kinase (Skav224893), showing 70% sequence similarity to rice PSTOL1, whose overexpression significantly enhances rice productivity under low phosphorus conditions [43], was expressed at a nearly 30 fold higher level under P stress. Gene homologs of *mei2*, a meiosis associated gene in yeast, were sharply down-regulated. Some expanded gene families reported in the genome were also detected as DEGs, such as P-stress responsive up-regulated AR repeat genes (Appendix A).

### 3.4. Functional Distribution of DEGs Responding to DOP Replacement

When the preferable DIP in the growth medium was replaced by the same molar concentration of glycerophosphate (Gro3P, DOP), the transcriptionally responsive DEG set in *F. kawagutii* changed. In total, 671 (8.05%) DEGs were identified, which composed 580 up- and 91 down-regulated genes (Table 2). Of all the 671 DEGs, 347 matched a functionally annotated gene in the databank (Appendix A). A total of eight genes were related to P utilization and exhibited higher expression levels under Gro3P as a P-source than DIP (Figure 4). Among them were two genes annotated as acid phosphatase that are believed to facilitate utilization DOP under DIP deficiency in plants [44]. Photosynthesis related genes were also markedly regulated by DOP. These included 17 DEGs of PsaD and PsaL in PSI, ferredoxin and cytochrome c6 in the photosynthetic electron transfer chain, PSII cytochrome b559, and chlorophyll a-c binding protein, as well as the chlorophyll synthesis enzyme, protochlorophyllide reductase. All PS related genes were promoted except cytochrome b559, which was down-regulated. 

The chemical defense genes that were homologs of gene families thought to protect against chemical stressors [25] were strongly induced by organophosphate replacement (Figure 4). These included the efflux pump ABC transporter and major facilitator superfamily, oxidative proteins, cytochrome P450 and flavin-containing monooxygenase, conjugative enzyme sulfotransferase, and the antioxidant proteins, manganese/iron superoxide dismutase. Nine heat shock proteins, including DnaJ, Hsp70, and Hsp90, also responded to DOP replacement for DIP. In addition, GO enrichment analysis showed that four categories were significantly impacted by the DOP replacement, including 22 DEGs specifying integral components of the membrane, 17 DEGs regulating metal ion binding, 4 DEGs involved in the tricarboxylic acid cycle (TCA cycle), and 12 DEGs associated with the carbohydrate metabolic process (Figure 4). Specially, two glyceraldehyde 3-phosphate dehydrogenase genes were greatly up-regulated in the DOP treatment (Appendix A). The expanded gene families identified as P- stress inducible above were also significantly regulated under DOP (Appendix A).

### 3.5. Comparison of DEGs between P Stress and DOP Replacement

The functional diversity of genes regulated by P stress was almost the same as that by DOP replacement, but the number of responsive gene families was different between the two treatments. There were 207 DEGs commonly responsive to P stress and DOP replacement in comparison to the control. Besides, 189 and 464 DEGs were unique to P stress and DOP replacement, respectively (Figure 5A). Overall, many photosynthesis genes were affected by both P conditions, although the impacted components were different. To uncover gene expression differences in *F. kawagutii* between P stress and DOP replacement, a KEGG pathway enrichment (*p*-value cutoff = 0.5) comparison on DEGs was performed. As Figure 5B shows, P stress and DOP replacement induced 88 and 137 orthologs of DEGs, respectively, involving 11 shared pathways in both groups. The common pathways included dominant lysosome, ubiquitin mediated proteolysis, oocyte meiosis, protein processing in the endoplasmic reticulum, photosynthesis, and the TCA cycle, among others. For the oocyte meiosis pathway, genes annotated as a best match in the InterPro databases of S-phase kinase-associated protein, serine/threonine-protein kinase, 14-3-3 proteins, and Poly (ADP-ribose) polymerase were up-regulated under both P treatments. Pathways specifically enriched under P stress included the cell cycle, cytochrome P450, TGF-beta signaling pathway, pentose and glucoronate interconversions, antenna proteins, arginine and proline metabolism, plant hormone signal transduction, ascorbate and aldarate metabolism, hippo signaling pathway, and apoptosis. In contrast, DEGs enrichment of DOP replacement uniquely comprised 22 pathways, including carbon metabolism and fixation, RNA degradation, and longevity regulating pathway (Figure 5B). 

## 4. Discussion

Our transcriptome sequencing had a limited depth coverage and lacked biological replicates (samples from triplicated cultures were pooled for sequencing) due to the high sequencing cost before 2011 when this project was conducted under the Marine Microbial Eukaryote Transcriptome Sequencing Program (MMETSP) [45]. To minimize the chance of falsely identifying DEGs given this limitation, we used a robust as possible and conservative approach in data analysis. First, we simultaneously used edgeR (medium BCV = 0.2) [46] and NOIseq [31] and identified differentially expressed genes (DEGs) only when they were recognized as such with statistical significance by both algorithms. EdgeR has been widely used to deal with no-replicate samples in phytoplankton research [47,48,49], and the combined use with another program (NOIseq) was expected to strengthen the conservative nature of DEG identification. Furthermore, recognizing that due to the low depth coverage we likely missed low-expression genes, our interpretation of data is limited to highly expressed genes. In addition, sequence reads that matched multiple genes and genes with low read counts (<10 count per million) [50], which is attributed to either short genes expressed at low levels or genes with small fold changes [51,52], were discarded. Consequently, the detected DEGs in this study represent both highly expressed and significantly regulated genes in *F. kawagutii* under the conditions examined in this study. Our observation that heat shock protein genes were up-regulated under heat stress as expected provides validating evidence of the suitability of our analysis approach.

Given our conservative way of identifying DEGs, it is no surprise that a smaller set of DEGs (357) responding to heat stress was identified in our study than that reported previously by Gierz et al. [14], in which 1776 DEGs with a ≥2-fold change in expression were found in Symbiodiniaceae species exposed to 31 °C (compared to 24.5 °C as control), accounting for 7.51% of the transcriptome (~23,654 unique genes). However, our smaller DEGs datasets were consistent with the previous larger datasets in revealing stress responsive heat shock and chaperonin proteins, ubiquitin proteasome, and alterations in the carbohydrate metabolic process (Table 3). Our DEG analysis in addition revealed a diverse set of genes that were transcriptionally regulated under heat stress, including high-affinity iron permease and iron binding molecules, oxygen transporter (phytoglobin), and genes specifying nutrient transport activities and cell features. 

The results from the current study, including the DEGs and the commonly expressed genes identified, provide a new perspective and a number of previously unsuspected processes or molecular functions involved in the stress response of Symbiodiniaceae. This study is also the first to explore transcriptomic responses to P deprivation and replacement of DIP with DOP. The results also raise many new questions to be addressed in future research, which have high potential to lead to new insights into triggers and processes of coral bleaching and other stress symptoms. These constitute a valuable genomic resource for further inquiries into mechanisms by which Symbiodiniaceae species (and the corresponding coral host) respond to and resist environmental changes and stress. However, we would like to caution, despite the value of the discovery of the genes expressed under the different temperature and phosphorus nutrient conditions, that the quantitative gene expression dynamics and any hypotheses generated based on the data reported here should be rigorously examined in future experiments with biological replicates.

### 4.1. “Core” Genes and Responsive Gene Groups in F. kawagutii

A total of 221 genes exhibited similar expression levels among the treatments, thus considered a constitutive gene repertoire of *F. kawagutii* (Appendix A). We propose that this belongs to the core gene set of this species. Most of these genes function in catalytic activities and binding (Figure 1), which are essential for metabolism and growth in the organism. To date, only a few stably expressed housekeeping genes have been identified in Symbiodiniaceae, as candidates of reference genes with which to normalize gene expression in molecular studies [69,70]. The 221 stably and highly expressed core genes identified in this study add more candidates for reference genes in future gene expression studies on *F. kawagutii*. However, the complete core gene set very likely consist of many more stably expressed genes that have escaped detection in this study due to their low expression levels and our limited sequencing depth; therefore, the initial core gene set reported here will be a primer of a broader search for core genes in this and other Symbiodiniaceae species in the future.

In contrast, 1091 genes were found uniquely expressed under one growth condition or significantly differentially expressed between conditions (Appendix A), which we postulate as environmentally responsive genes (ERGs). Even though some of these “uniquely expressed” genes may also be expressed (at low levels) under other conditions examined in this study, our filtering criteria were set to reduce the likelihood. If any of these would also be expressed under other conditions not investigated in this study, it remains to be found out in the future. Of these putative ERGs, there were eight duplicated-gene families, five of which were among the expanded gene families previously identified in the genome [23]. These eight showed changes in the expression level in response to all the three treatments used in this study. These are likely stress responsive genes (SRGs) in *F. kawagutii*, and potential biomarkers of stress for this species. These SRG gene families encode PPR (pentatricopeptide repeat), AR (ankyrin repeat), F-box, and FNIP repeat containing proteins, EF-hand calcium-binding protein, ZnF (Zinc finger) proteins, RCC1 (regulator of chromosome condensation), glycoside hydrolase, and glycosyl transferase, the functions of which will be discussed further in the following sections. 

PPR proteins are organelle RNA-binding proteins that mediate gene expression at the post-transcriptional level [71,72]. The AR domain typically mediates numerous protein–protein interactions [73], and contributes to various cellular functions, such as cell–cell signaling, cell–cycle regulation, and transport [74]. F-box and FNIP repeat-containing proteins and AR repeat proteins have been reported to play roles in degrading proteins through protein–protein interactions in thermal sensitive type C1 Symbiodiniaceae [75]. EF-hand calcium binding proteins influence many Ca^2+^-dependent cellular processes [76,77], such as cytoplasmic Ca^2+^ buffering, signal transduction protein phosphorylation, and enzyme activities [78,79]. Similarly, the diverse ZnF proteins serve as interactors of DNA, RNA, proteins, and small molecules [80]. GTP binding proteins of RCC1 are involved in cell cycle control and cellular processes [81,82], and provide a possible molecular basis for permanently condensed chromatin in dinoflagellates [83,84]. Glycoside hydrolases is known to affect cell wall architecture [85,86]; therefore, its up-regulation under stress suggests a role in stress adaptation. Glycosyl transferases catalyze the transfer of sugar residues [87]. Generally, it seems that these SRGs mostly function through molecular (protein–protein/RNA/DNA) interactions, probably rendering *F. kawagutii* better adapted to environmental changes and stresses.

### 4.2. Genes and Encoded Functions Responsive to Heat Stress in F. kawagutii

Under heat stress, the most remarkable transcriptomic response included the up-regulation of a zinc finger protein (ZnF, Skav215618) and down-regulation of an ABC transporter G family member (Skav232797). Overexpression of ZnF in transgenic *Arabidopsis* conferred improved thermal stress tolerance [88]. The ABCG transporters have been reported to be involved in biotic and/or abiotic stress responses [89,90,91,92,93]. The well-studied heat stress responsive genes coding for heat shock proteins [94,95,96] and glutathione S-transferase [97] were also up-regulated, while genes encoding conjugative proteins were down-regulated under thermal stress. Heat shock protein gene up-regulation, however, was also observed under P stress, indicating that these are likely broad stress responses, rather than specific heat stress response genes. Although heat stress has been implicated more in PSII damage [98,99], our finding of a down-regulation of PSI reaction center subunit IV suggests that PSI in *F. kawagutii* is also susceptible to heat stress. 

Heat stress-induced elevated expression of nutrient (formate/nitrite transporter, amino acid, ion transport protein, and p-type ATPase transporter) transporters suggests a higher nutrient demand and energy consumption under heat stress. In addition, two genes encoding plasma membrane permeases for high-affinity iron uptake (FTR1) were depressed under heat stress, suggesting decreased iron uptake. Meanwhile, iron binding activity was potentially promoted as inositol oxygenases and cytochrome c/P450 genes were significantly up-regulated, suggesting an increased iron demand under heat stress. *F. kawagutii* has been shown to have a higher iron requirement (500 pM soluble Fe for maximum growth rate) than other dinoflagellates [100,101], although it can maintain growth at low iron availability when other trace metals, such as manganese, copper, or zinc, are available [101]. All these in concert suggest that under heat stress, a fast transport system for iron (low-affinity) was functionally replacing the high-affinity iron transporter to meet the elevated demand for iron.

Genes related to the regulation of cell features, such as size and adhesion, were impacted under heat stress. Our results showed that genes encoding choline transporter-like protein 1 (*CTL1*), which regulates intracellular trafficking of auxin (a plant hormone) transporters to control seeding growth in *Arabidopsis* [102], were down-regulated under heat stress. Meanwhile, heat stress caused up-regulation of cytokinin riboside 5’-monophosphate phosphoribohydrolase, an enzyme activating cytokinin, another plant hormone that regulates cell division and differentiation. Whether the up-regulation of this cytokinin activating enzyme gene and down-regulation of *CTL1* gene would drive the cell cycle into arrest or promote cell division needs to be further investigated. Furthermore, a highly expressed gene encoding protocadherin fat (*Ft*) (FPKM = 221-854) was strongly up-regulated under thermal stress (log2[FC] = 2). In Drosophila, *Ft*, an atypical cadherin, regulates the Hippo pathway and plays a key role in regulating the organ size [103,104]. Also, cadherin is responsible for cell adhesion affected by cell Ca^2+^ homeostasis, and its up-regulation is supposed to prevent apoptosis in endosymbiosis [105]. It would be of interest to further explore the relationship between changes in cell size and adhesion as a potential adaptive mechanism to environmental stresses and *Ft* expression in Symbiodiniaceae. The detection of five up-regulated glycoside hydrolase genes provides evidence of cell wall changes under thermal stress because these genes are potentially involved in modulating the cell wall architecture [85,86]. Taken together, the results discussed above suggest that *F. kawagutii* copes with heat stress through gene regulation on cell features, including the cell cycle, adhesion, cell wall architecture, and physiological changes. 

Compared to previous findings (Table 3), the results reported here suggest that *F. kawagutii* uses unique mechanisms to cope with heat stress. Consistent with some previous studies [58], we found that photosystem II repair protein D1 gene was not among the up-regulated genes found under heat stress. It is tempting to speculate that the unique transcriptomic response (e.g., apparently elevated demand for iron, nutrients, and oxygen) may confer thermal tolerance, but this requires further investigation comparing *F. kawagutii* with known heat susceptible and resistant strains on multiple physiological as well as molecular parameters. 

### 4.3. Genes and Encoded Functions Responsive to P Deprivation in F. kawagutii

As an algae living in tropical low-nutrient environments, nutrient deficiency is potentially an important factor influencing photosynthesis and population growth. We examined how phosphorus (P) deprivation might cause transcriptomic responses. Under P deprivation, the most remarkable transcriptomic response included up-regulation of reticulocyte-binding protein 2 (Skav201252) and down-regulation of photosystem I reaction center subunit IV (Skav209900). Reticulocyte-binding protein 2 is known to be involved in reticulocyte adhesion (cell–cell adhesion) in the parasite *Plasmodium falciparum* [106]. Its role in *F. kawagutii* in responding to P deprivation is unclear and warrants further research in the future. 

From our data, we observed the up-regulation of alkaline phosphatase and down-regulation of metal-dependent phosphohydrolase under P deprivation, which suggest opposite functions of these enzymes in P metabolism. Alkaline phosphatase (AP) is widely known as being inducible by P stress in phytoplankton to facilitate the utilization of a phosphomonoester type of DOP [16]. Dinoflagellate AP belongs to an atypical PhoA group, the other members include homologs from diatoms, haptophytes, and other eukaryotic phytoplankton [107]. While different types of AP may require different cations as the dual-cofactors (e.g., Fe-Mg, Ca-Mg), PhoA is known to contain Zn-Mg as cofactors [107]. Consistently, AP has been reported to be Z limited in various phytoplankton groups [108,109]. The function of metal-dependent phosphohydrolase is less understood. In the dinoflagellate, *Prorocentrum donghaiense*, metal-dependent phosphohydrolase protein has been shown to be differentially expressed between the cell cycle phases using quantitative proteomic analysis [110], suggesting a role of the enzyme in phosphorylation-desphosphorylation of cell cycle regulating proteins. In the roots of the land plant model, *Arabidopsis thaliana*, it has been shown to be down-regulated after iron deprivation [111]. 

Photosynthetic capacity decreases as P deficiency stress increases, as demonstrated in plants [99]. Perhaps as a negative feedback to the decrease in photosynthetic capacity in *F. kawagutii*, the most abundant DEGs under P deprivation, occurred in the photosystem. Among them, proteins of ferredoxin (up) and flavodoxin (down), electron transporters, showed opposite regulation under P deprivation (Figure 4). They can replace each other in the photosynthetic electron transfer chain of cyanobacteria and algae [112]. In these photosynthetic taxa, flavodoxin is induced by iron deficiency while the iron-containing ferredoxin is down-regulated [39,40,41,42]. The up-regulation of flavodoxin might reflect lower iron demands under P stress. Although not investigated in this study, co-limitation of P and trace metals, such as iron and zinc, should be studied in the future.

DEGs involved in the production of chemical energy were all up-regulated, suggesting energy deficiency under P stress. DEGs enriched in oxidoreductase activity and heme binding were probably involved in P stress. The increased expression of abiotic defense genes or defensome (efflux, oxdative and antioxidant protein genes, and heat shock proteins) suggests these genes are also P stress responsive. 

Furthermore, we observed suppressed expression of two variants of the meiosis associated gene, *MEI2*, under P deprivation. *MEI2* is a RNA-binding protein involved in meiosis, crucial for commitment to meiosis (i.e., switching from mitotic to meiotic cell cycle) in the fission yeast, *Schizosaccharomyces pomb*, in which this gene is known to be induced by N-nutrient starvation [113,114,115]. The opposing response of this gene to N (in yeast) and P deprivation (in Symbiodiniaceae) is interesting, suggesting that P deficiency might inhibit meiosis whereas N starvation may induce it, thus leading to encystment. These might indicate that under P stress, *F. kawagutii* meiosis was repressed, which should be a topic of interest for future research.

### 4.4. Genes and Encoded Functions Are Responsive to DOP Replacement in F. kawagutii

When grown on Gro3P as the sole source of P, the most remarkable transcriptomic response in *F. kawagutii* included the up-regulation of reticulocyte-binding protein (Skav201252) and down-regulation of ATP-grasp fold (Skav221360) compared to the control (grown on phosphate). As it was also up-regulated under P deprivation, reticulocyte-binding protein up-regulation, key adhesins to recognize different receptors’ red blood cells in Plasmodium species [116,117], seems to be responsive to DIP deficiency. The implication of the response in the survival or physiology of *F. kawagutii* is unclear however. The ATP-grasp hold is one of the ATP-grasp superfamilies, which includes 17 groups of enzymes, catalyzing ATP-dependent ligation of a carboxylate containing molecule to an amino- or thiol-containing molecule [118], contributing to macromolecular synthesis. Its down-regulation under Gro3P treatment suggests a reduction of macromolecular synthesis when glycerophosphate is supplied as the sole P-source. This may represent an energy-saving benefit of utilizing DOP.

Compared to the DIP growth condition, acid phosphatases were uniquely up-regulated under Gro3P, suggesting its potential roles in DOP utilization. Furthermore, similar defensome sets as found under P deprivation were also observed under DOP, which in addition also induced the ABC transporter and major facilitator superfamily. The two membrane transporters couple solute movement to a source of energy [119], which potentially plays a vital role in transferring Gro3P. DnaJs play important roles in protein translation, folding, unfolding, translocation, and degradation, and regulates the activity of Hsp70s [120]. The abundant and up-regulated DnaJ as well as Hsp70 (Figure 4) were both induced by DOP condition in our study. The roles of *Hsp* genes in DOP metabolism remains to be further explored. 

The largest enriched GO term under the DOP condition was an integral component of the membrane, suggesting high transport activities to utilize DOP. It is interesting to note that the auxin efflux carrier was induced by organophosphate replacement, but not by P limitation. Auxin is a plant hormone, previously shown to occur and influence development in algae [121]. Auxin efflux carrier proteins influence many processes in plants, including the establishment of embryonic polarity, plant growth, apical hook formation in seedlings, and the photo- and gravitrophic responses [122,123,124]. In rice, the auxin efflux carrier gene is involved in the drought stress response [125]. In the genome of *F. kawagutii*, besides the four auxin efflux carrier genes, there are three auxin responsive GH3 genes, indicative of an auxin-based gene regulatory pathway in this species. It is unclear what physiological consequence the elevated expression of the auxin efflux carrier gene would lead to, but potentially it may be responsible for promoting cellular growth under the DOP condition. 

Utilization of Gro3P also induced differential gene expression related to the carbohydrate metabolic process and tricarboxylic acid cycle. Glyceraldehyde 3-phosphate dehydrogenase (*GAPDH*) and glycerol-3-phosphate dehydrogenase (*GPDH*) genes were greatly up-regulated. *GAPDH* interacts with different biomolecules, and has been known to play an important role in diatom’s ecological success [126]. In Chironomidae, *GAPDH* enhances heavy metal tolerance by adaptive molecular changes through binding at the active site [127]. *GPDH* is a very important enzyme in intermediary metabolism and as a component of the glycerophosphate shuttle it functions at the crossroads of glycolysis, oxidative phosphorylation, and fatty acid metabolism [128]. *GAPDH*- and *GPDH*-dependent metabolic pathways seem to be modulated by Gro3P utilization in *F. kawagutii*, and yet the physiological or ecological implications remain to be further investigated.

## 5. Conclusions

This study is the first to explore transcriptomic responses to P deprivation, DIP replacement with DOP, while also investigating thermal stress for the same genotype of Symbiodiniaceae, a strain that has been less frequently studied. The lack of biological replicates is a significant setback, making the data not as statistically solid as could be. With a dual-algorithm analysis strategy to ameliorate the impact of this limitation, we focused on the abundantly and stably expressed (core) genes of the species and a conservative set of differentially expressed genes (DEG) that responded to specific treatments. We identified 221 (2.04%) such core genes for *F. kawagutii*, which mostly were in the gene ontology terms of catalytic activity and binding. DEG results showed that eight duplicated gene families responded to all three treatments investigated in this study, including ankyrin repeat (AR)-, F-box and FNIP repeat- and pentatricopeptide repeat (PPR)- containing proteins, and EF-hand domain-containing calcium-binding proteins, zinc finger (ZnF) proteins, glycoside hydrolases, regulator of chromosome condensation (RCC1), and glycosyl transferases. These apparently are non-specific stress response genes in this species, mostly with predicted roles in active molecular (protein–protein/RNA/DNA) interaction. Specific to heat stress, 357 (4.42%) genes were found to be differentially expressed, apparently involved in cell wall modulation and the transport of iron, oxygen, and major nutrients, in addition to the expected up-regulation of heat shock protein genes. We did not observe a significant up-regulation of photosystem II repair protein D1 gene as expected under heat stress, which along with the distinct transcriptomic response observed suggests that this species has a unique mechanism by which to cope with heat stress and is possibly thermal tolerant. Our results also indicate that there is likely a higher demand for nutrients, iron, and oxygen under heat stress. About as many DEGs (396, 4.73%) were identified under P deprivation while nearly double of that (671, 8.05%) were detected under DOP (glycerophosphate) utilization; in both cases, most of the DEGs were up-regulated and predicted to function in the photosystem and defensome, indicating that photosynthesis and defense are probably the most markedly impacted physiologies under varying P-nutrient conditions. 

In sum, the data reveal a wide range of, and in many cases previously unrecognized, molecular mechanisms to cope with heat stress and phosphorus nutrient stress conditions. The regulation of these mechanisms may enable *F. kawagutii* to adapt to temperature and P-nutrient varying environments. This study provides novel insights into responses in *F. kawagutii* to both temperature and P nutrient variations, and with cautioning of the no-replicate limitation, sets a valuable framework for more transcriptomic research in the future on Symbiodiniaceae species to reliably (using biological replicates) uncover common and stress-specific features of stress sensitive (bleaching prone) and tolerant strains and to elucidate triggers of coral bleaching.

## Figures and Tables

**Figure 1 microorganisms-07-00096-f001:**
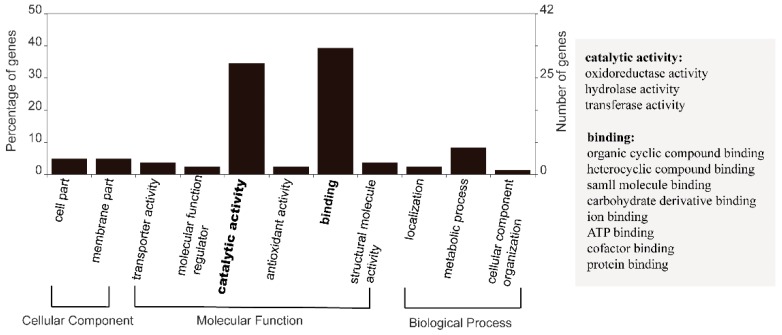
Core genes GO annotation category. Core genes were defined as genes commonly detected under all four conditions: control, heat stress, P deprivation, and DOP replacement, and showed expression of ≥50th percentile average FPKM and low CV of ≤0.1.

**Figure 2 microorganisms-07-00096-f002:**
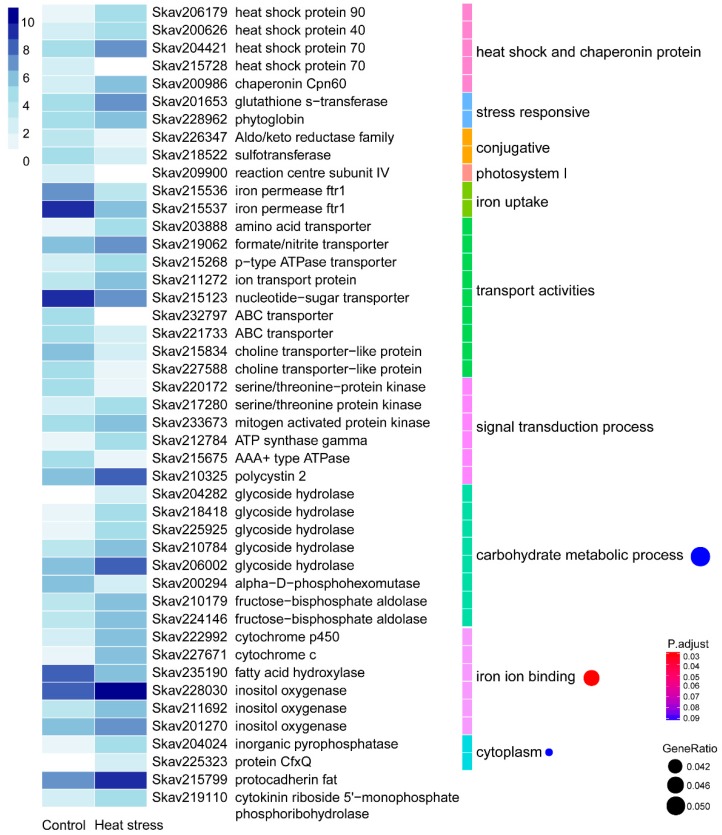
Differential gene expression in response to heat stress (30 °C). The heatmap color strength represents log2-transformed gene expression levels estimated as fragments per kilo-base of exon model per million mapped fragments (FPKM), from dark blue (highest), light blue, to white (lowest). Each color bar on the middle right marks a functional category. Each dot on the right marks a category based on GO enrichment (*p*-value cutoff = 0.1). The dot size represents the enriched DEGs count. The color strength represents the *p*-value. Control: SymkaSL2 grown in L1 medium at 25 °C. HS: SymkaSL1 was grown in L1 medium at 30 °C.

**Figure 3 microorganisms-07-00096-f003:**
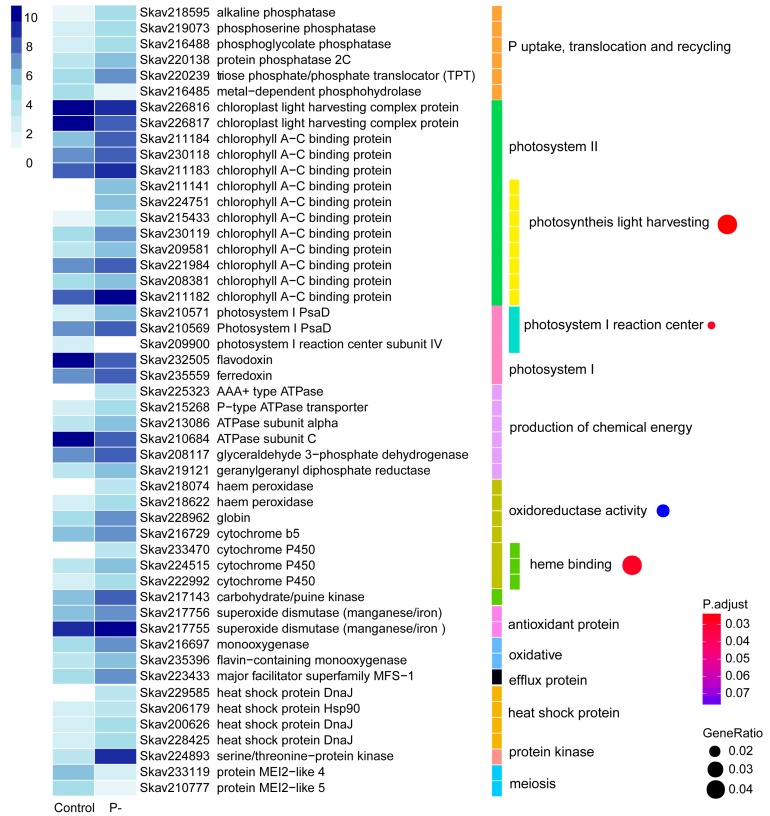
Differential gene expression in response to phosphate deprivation. The heatmap color strength represents log2-transformed gene expression by fragments per kilo-base of exon model per million mapped fragments (FPKM), from dark blue (highest), light blue, to white (lowest). Each color bar on the middle marks a functional category. Each dot on the right marks a category based on GO enrichment (*p*-value cutoff = 0.1). The dot size represents enriched DEGs count. The color strength represents the *p*-value. Control: SymkaSL2 grown in L1 medium at 25 °C. P-: SymkaSL3 was grown in L1 medium with depleted DIP at 25 °C.

**Figure 4 microorganisms-07-00096-f004:**
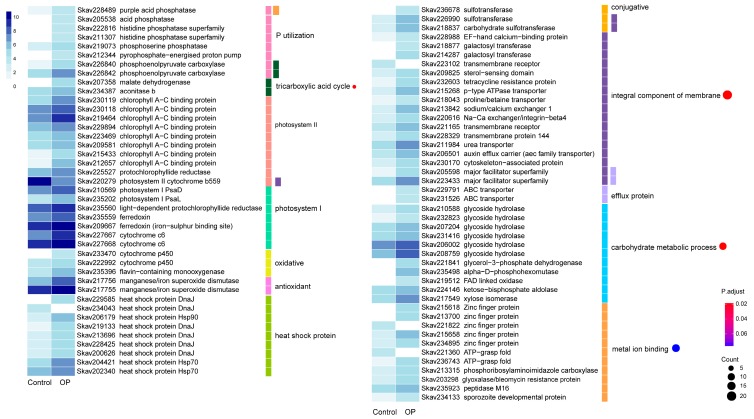
Differential gene expression in response to organophosphate. The heatmap color strength represents log2-transformed gene expression by fragments per kilo-base of exon model per million mapped fragments (FPKM), from dark blue (highest), light blue, to white (lowest). Each color bar on the middle right marks a functional category. Each dot marks a category based on GO enrichment (*p*-value cutoff = 0.1). The dot size represents the enriched DEGs count. The color strength represents the *p*-value. Control: SymkaSL2 was grown in L1 at 25 °C. OP: SymkaSL4 was grown in L1 medium with DIP replaced by glycerophosphate at 25 °C.

**Figure 5 microorganisms-07-00096-f005:**
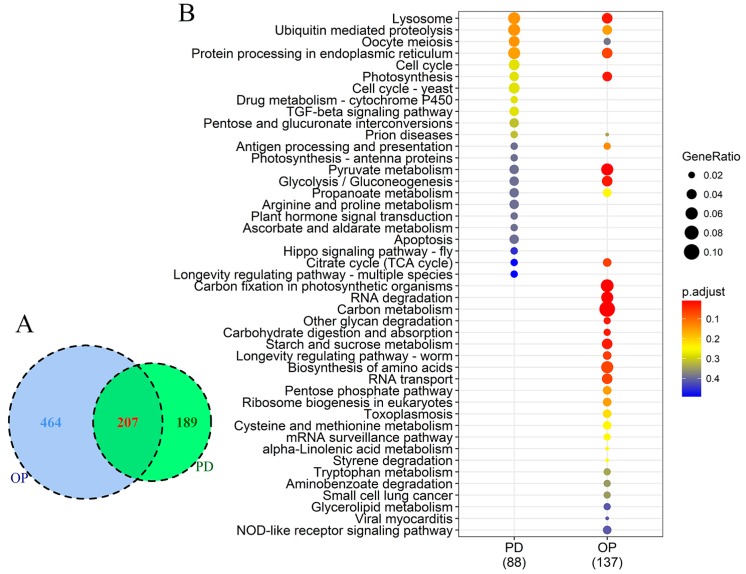
Comparison of differentially expressed genes between P-deprived (PD) and organophosphate (OP) conditions. (**A**) Venn diagram showing numbers of DEGs common in both or unique to each of the two P conditions. (**B**) KEGG pathway enrichment; dot size represents enriched DEGs count; color strength represents the *p*-value (from lowest in red to highest in blue).

**Table 1 microorganisms-07-00096-t001:** RNA-seq information of *F. kawagutii*.

Sample ID	MMETSP ID	SRA ID	Condition	Clean Data Size (Mbp)	Mapping Rate
SymkaSL1	MMETSP0132	SRR1300302	heat stress	660	59.21%
SymkaSL2	MMETSP0133	SRR1300303	normal	550	61.29%
SymkaSL3	MMETSP0134	SRR1300304	P deprivation	695	67.43%
SymkaSL4	MMETSP0135	SRR1300305	Gro3P replacement	1150	62.34%

Note: All samples were grown in L1 medium amended natural seawater in 25 °C except listed condition (see Methods).

**Table 2 microorganisms-07-00096-t002:** Number of DEGs under different conditions identified by NOIseq and edgeR.

Group	AEG	NOIseq	edgeR	NOIseq + edgeR
**HS**	8081	1108 (13.71%)	450 (5.57%)	357 (4.42%, 249↑+108↓)
**P-**	8364	1535 (18.35%)	557 (6.66%)	396 (4.73%, 332↑+64↓)
**DOP**	8335	4707 (56.47%)	1111 (13.33%)	671 (8.05%, 580↑+ 91↓)
**Total union**	10,857	5397	1601	1091

Note: AEG: actively expressed genes (average CPM ≥ 10). HS: heat stress; P-: P deprivation; DOP: dissolved organophosphate.

**Table 3 microorganisms-07-00096-t003:** Major findings of previous heat stress related studies on Symbiodiniaceae.

Clade/Type	Conditions (control; stress)	Major Findings	Reference
*Symbiodinium. microadriaticum*	Heat stress (26 °C; 20–36 °C)	Photosynthesis was impaired at temperatures above 30 °C and ceases completely at 34–36 °C.	Iglesias-Prieto et al. 1992 [53]
Symbiodiniaceae Clade C3	Warming (21.1 °C; 28.7 °C); Eutrophication (ammounium); increasing CO_2_ levels	Identified 1456 unique ESTs, among which 561 (44%) were functionally annotated. Most of them were related to posttranslational modification, protein turnover, and chaperones; energy production and conversion.	Leggat et al. 2007 [54]
Symbiodiniaceae OTcH-1 (Clade A) CS-7 (Clade A)	Heat stress (25–34 °C)	Inhibition of de novo synthesis of intrinsic light-harvesting antennae [chlorophyll a– chlorophyll c2–peridinin–protein complexes (acpPC); photoinhibition of photosystem II observed in CS-7 at 34 °C, but not in OTcH-1.	Takahashi et al. 2008 [55]
Symbiodiniaceae Type C1 Clade D	bleaching (28 °C; 30, 31, and 32 °C) heat stress (26 °C; 29, and 32 °C)	Lower metabolic costs and enhanced physiological tolerance of *Acropora tenuis* juveniles when hosting Symbiodiniaceae type C1 compared with type D.	Abrego et al. 2008 [56]
Symbiodiniaceae CCMP829 (Clade A)	Heat stress (27 °C; 34 °C)	Enhanced nitric oxide (NO) production at high temperatures.	Bouchard et al. 2008 [57]
Symbiodiniaceae OTcH-1 (Clade A) CS-73 (Clade A)	Heat stress (25°C; ~34 °C)	Thermal resistance is not associated with de novo synthesis of D1 protein.	Takahashi et al. 2009 [58]
Symbiodiniaceae Type C3	Heat stress (27 °C; 34 °C)	Expression of stress responsive and carbon metabolism genes were up-regulated in coral host, but seldom and with smaller fold changes in the symbiont, during the experimental bleaching event.	Leggat et al. 2011 [59]
Symbiodiniaceae CassKB8 (Clade A) Mf1.05b (Clade B)	Heat (27 °C; 30-31 °C); cold (27 °C; 19 °C); light (120 µmoL photons/m^2^/s); dark (darkness for 6 days)	Generated 56,000 assembled sequences per species; found a complete set of core histones, a low number of transcription factors (cold shock domain was predominant), and a high number of antioxidative genes.	Bayer et al. 2012 [60]
Symbiodiniaceae CCMP827 (Clade A) CCMP831 (Clade A) CCMP830 (Clade B) CCMP421 (Clade E)	Heat stress (25 °C; 30 °C, 35 °C)	Enhanced thermal tolerance of PSII at elevated temperatures.	Takahashi et al. 2013 [11]
Symbiodiniaceae Type D2 Type C3K	Heat stress (26.8–34.5 °C; 27–37.6 °C for 3 days)	No DEGs after heat stress within each type; Hundreds of DEGs after heat stress between the two types.	Barshis et al. 2014 [12]
Symbiodiniaceae Ap1(Clade B1) CCMP2466 (Clade C1) CCMP421 (Clade E) Mv (Clade F1)	Heat stress (25 °C; 29 °C, 33 °C)	In Symbiodiniaceae clades B1, C1, and E, declining photochemical efficiency (Fv /Fm) and death at 33 °C were generally associated with elevated superoxide dismutase (SOD) activity and a more oxidized glutathione pool. Clade F1 exhibited no decline in Fv /Fm or growth, but showed proportionally larger increases in ascorbate peroxidase (APX) activity and glutathione content (GSx), while maintaining GSx in a reduced state.	Krueger et al. 2014 [61]
Symbiodiniaceae Y106 (Clade A) K100 (Clade B) Y103 (Clade C) K111 (Clade D) K102 (Clade F)	Heat stress (25 °C; 33 °C)	Decreased growth rate and photosynthesis at elevated temperature in clades A and B, but not in clades D and F.	Karim et al. 2015 [13]
Symbiodiniaceae Type C3 Type C15	Heat stress (28 °C; 33 °C)	No significant changes in enzymatic antioxidant defense detected in the symbiont. Preceded significant declines in PSII photochemical efficiencies.	Krueger T et al. 2015 [62]
Symbiodiniaceae Clade C3	Heat stress (increasing daily from 25 °C to 34 °C)	At day 8, photochemical efficiency was decreased. On day 16, symbiont density was significantly lower. Three acpPC genes were up-regulated when temperatures above 31.5 °C.	Gierz et al. 2016 [63]
Symbiodiniaceae thermos-sensitive SM (Type C1) thermos-tolerant MI (Type C1)	Heat stress (27 °C; 32 °C)	After 9 days at 32 °C, the two populations showed no physiological stress, but the enhanced meiosis genes. After 13 days at 32 °C, SM population showed decreasing photochemical efficiency and increasing ROS, MI exhibited no physiological stress and enhanced expression of genes of ROS scavenging and molecular chaperone.	Levin et al. 2016 [64]
Symbiodiniaceae Clades A, B, D, F	Heat stress (25 °C; 32 °C)	Sixteen Symbiodiniaceae isolates were clustered into three novel functional groups based on their physiological response to heat stress: thermally tolerant, thermally susceptible and thermally.	Goyen et al. (2017) [65]
Symbiodiniaceae Clade F	Heat stress (24.5 °C; 31 °C for 28 days)	37.01% DEGs of the transcriptome (∼23,654 unique genes found at FDR < 0.05), with 92.49% DEGs at ≤2-fold change. The DEGs encoded stress response components, glyoxylate cycle enzymes, and altered metabolic processes.	Gierz et al. 2017 [14]
Symbiodiniaceae Type A3 Type B1 Type B2 Type C2 Type D1a Type F	Heat stress (26 °C; 20–33 °C)	Six Symbiodiniaceae genotypes showed significant differences in the response patterns under heat stress. While some types photosynthesized, respired, and grew at 33 °C, others showed a partial or complete inhibition.	Gregoire et al. 2017 [66]
Symbiodiniaceae CCMP2467 (Clade A)	Heat stress (26 °C; 36 °C) Cold stress (26 °C; 16 °C) Dark stress (no daybreak)	Verified the existence of heat stress-activated Ty1-copia-type LTR retrotransposons and its recent expansion events in the *S. microadriaticum*.	Chen et al. 2018 [67]
*Breviolum. minutum* (Clade B) *Cladocopium goreaui* (Clade C) *Durusdinium trenchii* (Clade D)	Heat stress (26 °C; 32 °C)	Heat stress inhibited cell cycle progression and arrested all strains in G1 phase.	Fujise et al. 2018 [68]
*Fugacium kawagutii* CCMP2468 (Clade F)	Heat stress (25 °C; 30 °C) P deprivation (25 °C; P-) DOP utilization (25 °C; DOP)	Documented 357 (4.42%) DEGs under heat stress putatively involved in molecular interaction, cell wall modulation and transport, in addition to heat shock proteins reported previously. Documented 396 (4.73%) DEGs under P deprivation, and 671 (8.05%) DEGs under DOP utilization, which have not been studied previously, and both groups of DEGs putatively function in photosystem and defensome.	This study

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
