# Peer review of "Transcriptomic Responses to Thermal Stress and Varied Phosphorus Conditions in Fugacium kawagutii"

_microorganisms, 2019, doi:10.3390/microorganisms7040096_

Round 1

Reviewer 1 Report

I think this is an interesting data analysis of the response of an important clade of coral symbionts (Fugacium) in relation to thermal stress and nutrient depletion.

In general, the comparisons are interesting and some relevant pathways and genes are found that may point to future experiments.

However, this study has a major problem, which is  that there is no biological replication. This seems to be glossed over in the manuscript, and is only briefly mentioned in the methods, and not stated at all in the Abstract, Results, Discussion or Conclusion. I think that given that there is no biological replication, the data can only be interpreted as an exploratory data analysis showing some interesting trends, rather than a study with statistically relevant results. Therefore, I recommend major revisions in order for the authors to remove the references to the statistical significance of results, and to state in the Abstract, Discussion and Conclusion that there was no biological replication, and that these results are exploratory.

Methods:

Section 2.1: I think its important to state here that the biological replicates were pooled before RNA extraction. This has a strong influence on the interpretation of results, so its important to state it clearly here.

Section 2.5 – I think its not really accurate to state that ‘reliable differential gene expression profiles’ can be generated in the absence of biological replicates. Statistically, it is not possible in an ecological study to calculate a significant difference between samples where there is no replication.  It is true to say that there are software that attempt to allow for some comparisons between samples in the absence of replication. However, I think these need to be considered as exploratory data analysis methods to indicate tendencies, rather than an actual differential gene expression analysis.  For example, in the documentation for EdgeR, the following is written on Pg 22:

  edgeR is primarily intended for use with data including biological replication. Nevertheless,

RNA-Seq and ChIP-Seq are still expensive technologies, so it sometimes happens that only

one library can be created for each treatment condition. In these cases there are no replicate

libraries from which to estimate biological variability. In this situation, the data analyst is

faced with the following choices, none of which are ideal. We do not recommend any of

these choices as a satisfactory alternative for biological replication. Rather, they are the best

that can be done at the analysis stage, and options 2–4 may be better than assuming that

biological variability is absent.

1. Be satisfied with a descriptive analysis, that might include an MDS plot and an analysis

of fold changes. Do not attempt a significance analysis. This may be the best advice. “

 Results:

Given the above, I think that the statistical analyses of the results needs to be removed, and it needs to be rewritten as a descriptive paper, including some fold changes, but no attempts at statistical analyses. I think attempting statistical analyses is misleading, and could confuse other authors who would then equate the statistical values with true probabilities that might be derived from a study that did use real biological replication. As a descriptive study or exploratory data analysis, this study would still be relevant and very useful, as this is an interesting organism and information about its response to temperature and nutrient treatments is required.

Author Response

Response to Reviewer 1 Comments

Point 1: I think this is an interesting data analysis of the response of an important clade of coral symbionts (Fugacium) in relation to thermal stress and nutrient depletion. In general, the comparisons are interesting and some relevant pathways and genes are found that may point to future experiments. However, this study has a major problem, which is that there is no biological replication. This seems to be glossed over in the manuscript, and is only briefly mentioned in the methods, and not stated at all in the Abstract, Results, Discussion or Conclusion. I think that given that there is no biological replication, the data can only be interpreted as an exploratory data analysis showing some interesting trends, rather than a study with statistically relevant results. Therefore, I recommend major revisions in order for the authors to remove the references to the statistical significance of results, and to state in the Abstract, Discussion and Conclusion that there was no biological replication, and that these results are exploratory.

Response 1: We would like to clarify we did make clear about the lack of biological replicates in Discussion (in fact in the very beginning of Discussion), we caution the authors adequately and Conclusion (…ameliorate impact of …limitation) as well as Methods. But now we agree with the reviewer that we should have made it as clear in Abstract and emphasize this weakness in Results as well. We now have added this in the two parts, and also have given more emphasis of the limitation and cautioning in Discussion and Conclusion and wherever we see relevant. Please see line 19, lines 166-168, lines, 340, 342, 346-349, and 566-568.

Point 2: Section 2.1: I think it’s important to state here that the biological replicates were pooled before RNA extraction. This has a strong influence on the interpretation of results, so it’s important to state it clearly here.

Response 2: Many thanks for your suggestion, which we totally agreed. Indeed, we did state it in this section: in the second last sentence (line 106): “Before sequencing, RNA extracts from the triplicate cultures were pooled at equal RNA quantity”.

Point 3: Section 2.5 – I think its not really accurate to state that ‘reliable differential gene expression profiles’ can be generated in the absence of biological replicates. Statistically, it is not possible in an ecological study to calculate a significant difference between samples where there is no replication. It is true to say that there are software that attempt to allow for some comparisons between samples in the absence of replication. However, I think these need to be considered as exploratory data analysis methods to indicate tendencies, rather than an actual differential gene expression analysis. For example, in the documentation for EdgeR, the following is written on Pg 22: “edgeR is primarily intended for use with data including biological replication. Nevertheless, RNA-Seq and ChIP-Seq are still expensive technologies, so it sometimes happens that only one library can be created for each treatment condition. In these cases there are no replicate libraries from which to estimate biological variability. In this situation, the data analyst is faced with the following choices, none of which are ideal. We do not recommend any of these choices as a satisfactory alternative for biological replication. Rather, they are the best that can be done at the analysis stage, and options 2–4 may be better than assuming that biological variability is absent.

Response 3: This inaccurate wording has been removed in this sentence. See line 134. However, even though in absence of replication, we carefully select the stricter biological coefficient of variation (BCV) of 0.2 to estimate dispersion in edgeR. The edgeR was used in many studies in absence of replication, for example the following: (1) Gong W, Browne J, Hall N, Schruth D, Paerl H, Marchetti A: Molecular insights into a dinoflagellate bloom. The Isme Journal 2016, 11:439. (2) Shi X, Lin X, Li L, Li M, Palenik B, Lin S: Transcriptomic and microRNAomic profiling reveals multi-faceted mechanisms to cope with phosphate stress in a dinoflagellate. The Isme Journal 2017, 11:2209. (3) Harke MJ, Juhl AR, Haley ST, Alexander H, Dyhrman ST: Conserved Transcriptional Responses to Nutrient Stress in Bloom-Forming Algae. Frontiers in Microbiology 2017, 8(1279). Additionally, we use another one more method (NOIseq) accounting for biological variation to better identify DEGs. We have added this information in the first paragraph of Discussion.

Point 4: 1. Be satisfied with a descriptive analysis, that might include an MDS plot and an analysis of fold changes. Do not attempt a significance analysis. This may be the best advice. “

Response 4: Thanks for your suggestion. As we explained above, we have now made very clear the setback of the non-replicate sequencing results, and caution readers about the quantitative gene expression dynamics. We think it is important to describe that what we present in the paper has been rigorously filtered through two methods and we did not simply present every gene that showed expression difference between conditions. We also think that readers, with our repeated cautioning the paper, are well informed and can make their own judgement and can compare future studies with the reported data here.

Point 5: Results: Given the above, I think that the statistical analyses of the results needs to be removed, and it needs to be rewritten as a descriptive paper, including some fold changes, but no attempts at statistical analyses. I think attempting statistical analyses is misleading, and could confuse other authors who would then equate the statistical values with true probabilities that might be derived from a study that did use real biological replication. As a descriptive study or exploratory data analysis, this study would still be relevant and very useful, as this is an interesting organism and information about its response to temperature and nutrient treatments is required.

Response 5: Thanks for your suggestion. As we explained above, we think we have taken necessary step to be conservative and to be cautioning. Our analysis is more strict then many two or three years ago when authors used edgeR alone to analyze statistical analysis on differentially expressed genes. The statistical analysis comes automatically with any DEG expression analysis, and this is the best way we can think of analyzing the data.

Reviewer 2 Report

Despite the lack of coverage, the transciptome data would be a welcome addition to the present pool of responsive effectof this ecologically important dinoflagellates to stresses

Can the discussion in Table 3, which is the significant result and interpretation of the present data,  be spelt out the differential results obtained in the present study;

Whereas the common genomic data e.g. Rccs, rna-binding domain etc,  could be summarized in a supplementary table

The manuscript set out to address the followings:

“F. kawagutii belongs to clade F and was originally isolated from scleractinian coral Montipora verrucosa in Hawaii where ambient temperature is about 25°C [18],  but this specific genotype has only been occasionally found in subsequent studies (in Pocillopora  damicornis in Heron Island [19] and in Hong Kong [Lin unpublished data], both in the Pacific). This  raises a question if this is like the thermal resistant clade D genotypes that are rare under normal  conditions but resistant to stress due to physiological tradeoff [20-22].

This was not addressed in the results nor the discussion

Any comments correlating to in hospice situation would lead interested readers to specific parts of the results

I don’t have time to read through the English but

 “Fig.1. insert should “protine-binding” be “protein-binding”

Author Response

Response to Reviewer 2 Comments

Point 1: Despite the lack of coverage, the transciptome data would be a welcome addition to the present pool of responsive effect of this ecologically important dinoflagellates to stresses. Can the discussion in Table 3, which is the significant result and interpretation of the present data, be spelt out the differential results obtained in the present study.

Response 1: Thanks for the encouraging comments. Regarding Table 3, we now have added some wording to highlight what is new finding in the present study. Due to space limit, we cannot elaborate too much, however.

Point 2: Whereas the common genomic data e.g. Rccs, rna-binding domain etc, could be summarized in a supplementary table

Response 2: Thanks for your suggestion. The common genomic data in our study refers to the expanded gene families reported in genome data. We marked them in the supplementary table S3-S5 in column J with “duplicated gene families (expanded).”

Point 3: The manuscript set out to address the followings: “F. kawagutii belongs to clade F and was originally isolated from scleractinian coral Montipora verrucosa in Hawaii where ambient temperature is about 25°C [18], but this specific genotype has only been occasionally found in subsequent studies (in Pocillopora damicornis in Heron Island [19] and in Hong Kong [Lin unpublished data], both in the Pacific). This raises a question if this is like the thermal resistant clade D genotypes that are rare under normal conditions but resistant to stress due to physiological tradeoff [20-22].” This was not addressed in the results nor the discussion. Any comments correlating to in hospice situation would lead interested readers to specific parts of the results

Response 3: Thanks for your valuable comments. In Discussion (section 4.2), we have tried to address this to the extent possible given the limitation of our dataset. We suggested that the unique transcriptomic response (e.g. apparently elevated uptake of iron, nutrients, and oxygen, modulation of cell wall and other cell features) might confer thermal tolerance but this requires further investigation comparing F. kawagutii with known heat susceptible and resistant strains on multiple physiological as well as molecular parameters. Please see particularly the last paragraph of section 4.2

Point 4: I don’t have time to read through the English, but“Fig.1. insert should “protine-binding” be “protein-binding”

Response 4: Thanks for your comments. The word in Fig. 1 has been revised. We also modified the Fig. 1 for better display. In addition, we carefully revised the English writing according to the comments provided by other reviewers. 

Reviewer 3 Report

I congratulate the authors on a nice study.  The lack of replication while understandable because of cost does unfortunately detract from the product.  The highly conservative approach (2 tests) is sound, but I also think it would be useful to see what confidence level is needed to approach what Giertz et al. 2017 reported.

A copy of the edited MS is included- notable issues are typos in the literature cited, and some suggestions for cumbersome English constructions. 

Author Response

Response to Reviewer 3 Comments

Point 1: I congratulate the authors on a nice study. The lack of replication while understandable because of cost does unfortunately detract from the product. The highly conservative approach (2 tests) is sound, but I also think it would be useful to see what confidence level is needed to approach what Giertz et al. 2017 reported.

Response 1: We appreciate your constructive comments. We found stress response (e.g. GST, heat shock proteins) and photosynthesis related genes (e.g. Photosystem I reaction center subunit IV) were regulated in our study, which were similar to Giertz et al. 2017 reported. We did not find other DEGs reported in Giertz et al. study, which might be because we used the very conservative approach of analysis and as a consequence yielded a much smaller set of DEGs (357) identified than that reported by Gierz et al. (1,776 DEGs). However, we conducted GO enrichment based on DEGs to figure out important pathways affected by heat stress (e.g. iron ion binding, carbohydrate metabolic process), which were not reported in Giertz et al. 2017 study.

Point 2: A copy of the edited MS is included- notable issues are typos in the literature cited, and some suggestions for cumbersome English constructions.

Response 2: Thanks for your suggestions. According to your comments, the references and English constructions have been carefully revised. Please see in the revised manuscript.